# Biologicalization of Smart Manufacturing Using DNA-Based Computing

**DOI:** 10.3390/biomimetics8080620

**Published:** 2023-12-18

**Authors:** Sharifu Ura, Lubna Zaman

**Affiliations:** 1Division of Mechanical and Electrical Engineering, Kitami Institute of Technology, 165 Koen-cho, Kitami 090-0055, Japan; 2Advanced Manufacturing Engineering Laboratory, Kitami Institute of Technology, 165 Koen-cho, Kitami 090-0055, Japan; lubnazaman30@gmail.com

**Keywords:** central dogma of molecular biology, DNA-based computing, smart manufacturing, biologicalization of manufacturing, image processing, pattern recognition, sensor signals

## Abstract

Smart manufacturing needs cognitive computing methods to make the relevant systems more intelligent and autonomous. In this respect, bio-inspired cognitive computing methods (i.e., biologicalization) can play a vital role. This article is written from this perspective. In particular, this article provides a general overview of the bio-inspired computing method called DNA-Based Computing (DBC), including its theory and applications. The main theme of DBC is the central dogma of molecular biology (once information of DNA/RNA has got into a protein, it cannot get out again), i.e., DNA to RNA (sequences of four types of nucleotides) and DNA/RNA to protein (sequence of twenty types of amino acids) are allowed, but not the reverse ones. Thus, DBC transfers few-element information (DNA/RAN-like) to many-element information (protein-like). This characteristic of DBC can help to solve cognitive problems (e.g., pattern recognition). DBC can take many forms; this article elucidates two main forms, denoted as DBC-1 and DBC-2. Using arbitrary numerical examples, we demonstrate that DBC-1 can solve various cognitive problems, e.g., “similarity indexing between seemingly different but inherently identical objects” and “recognizing regions of an image separated by a complex boundary.” In addition, using an arbitrary numerical example, we demonstrate that DBC-2 can solve the following cognitive problem: “pattern recognition when the relevant information is insufficient.” The remarkable thing is that smart manufacturing-based systems (e.g., digital twins and big data analytics) must solve the abovementioned problems to make the manufacturing enablers (e.g., machine tools and monitoring systems) more self-reliant and autonomous. Consequently, DBC can improve the cognitive problem-solving ability of smart manufacturing-relevant systems and enrich their biologicalization.

## 1. Introduction

Manufacturing adds value to an economy. One of the acclaimed manufacturing concepts is smart manufacturing [1,2] or Industry 4.0 [3,4]. Smart manufacturing employs information and communication technology-based systems to solve manufacturing problems. Among others, cyber-physical systems, Industrial Internet of Things, big data, big data analytics, artificial intelligence, machine learning, digital models, digital shadows, digital twins, sensor signaling, virtual reality, and digital manufacturing commons are the main constituents of smart manufacturing [1,2,3,4,5]. These constituents are embedded into manufacturing enablers (machine tools, human resources, peripherical equipment, computer-aided design, manufacturing and process planning systems, enterprise resources planning systems, and supply chain systems) [1,2,3,4,5]. The goal is to make the manufacturing enablers more self-reliant or autonomous [2,4,5]. Consequently, the above-mentioned constitutes and enablers need to perform some cognitive tasks (pattern recognition, knowledge elicitation, adaptation to emerging environments, and dealing with uncertainty) [2,4,5]. In this case, different aspects of biological systems can be used as a source of motivation, as suggested by the “biologicalisation” movement of manufacturing [6,7]. (This article uses the term “biologicalization” instead of “biologicalisation” to keep spelling consistency.) In biologicalization, three facets are considered: (1) bio-inspiration, (2) bio-integration, and (3) bio-intelligence [6,7]. Bio-inspiration refers to a method that is related to a biological phenomenon. Among others, genetic algorithms/programming [8,9,10] and artificial neural networks/deep learning [11] are the prominent bio-inspired computing methods. Genetic algorithm/programming is based on the biological phenomenon called evolution [8,9,10], whereas artificial neural network/deep learning is based on neuron signal processing that takes place in the brain of a biological organism [11]. Design and manufacturing scientists have extensively been using these computing methods to solve complex problems (see [12] for some typical examples). On the other hand, bio-integration is a relatively new addition to the biologicalization of manufacturing. It integrates biotechnological approaches to solve manufacturing problems. One of the remarkable examples is developing more environment- and human-health-friendly cutting fluids containing microalgae species as a lubricant component (see [13] for more details). Finally, bio-intelligence refers to imparting human-like multiple problem-solving capacity to a manufacturing system. For example, a bio-intelligence-based design system must be capable of designing products by translating customer needs into product specifications. This facet perhaps indicates artificial general intelligence [14,15], the next frontier of intelligent systems development per se. Thus, among the three facets of biologicalization, bio-intelligence is the most challenging because it needs knowledge creation as well as simultaneous management of all four types of knowledge rather than the mere use of existing knowledge due to deduction or induction (see [16] for more details).

Nevertheless, biologicalization in manufacturing can be traced back to the initiative of Ueda and others in Japan called Biological/Bionic Manufacturing Systems (BMSs) [17,18,19,20,21,22]. The BMS acknowledges the ever-changing external-internal environments of the life cycle of a product and lets the enabling systems self-grow, self-organize, adapt, and evolve [17,18,19,20,21,22]. This attempt to articulate manufacturing systems leads to other concepts of manufacturing system modeling, namely, holonic and fractal manufacturing systems [23,24,25], where the main target is to take advantage of object-oriented programming-driven software systems. However, for upholding the self-growing, self-organizing, adaptive, and evolving characteristics of a manufacturing system, two types of information are recommended: (1) “DNA-type information” and “BN (Brain and Neurons)-type information” [17,18,22]. As far as biological systems are concerned, DNA-type information means inherited information (i.e., genetic information (DNA) that is passed from one cell to another and even one generation to another). And, BN-type information means learned-type information (i.e., the information that the brain learns using the neural network). For the BMS, on the other hand, DNA-type information is a metaphor. It refers to information that a manufacturing system must inherit while solving product life cycle-relevant problems. Similarly, BN-type information is also a metaphor for the BMS. It means the information that the system must learn while solving product life cycle-relevant problems. One of the main concerns of BMS is what kind of information out of DNA-type information and BN-type information should be prioritized. The answer is that DNA-type information is more important than BN-type information because superior biological functions, such as flexibility, autonomy, self-formation, and self-recovery, are expressed by DNA-type information rather than BN-type information (see [17,18,22] for more details). The remarkable thing is that an integrated approach called “gentelligent” manufacturing systems has been developed to gain benefit from DNA-type information and BN-type information [26,27].

However, some authors [28,29,30,31,32,33,34,35] addressed the biologicalization movement of manufacturing based on the *central dogma of molecular biology* [36,37,38] introduced by Crick (one of the Nobel laureates who discovered the double helix structure of DNA). Crick stated, “Once information has got into a protein, it cannot get out again. Information here means the sequence of the amino acid residues, or other sequences related to it” [36,37,38]. This means genetic information can be passed to protein (sequence of amino acids) from DNA or RNA (sequence of nucleic acids), but the reverse is not possible. Therefore, biological systems only allow “DAN to DNA”, “DNA to RNA”, and “DNA to protein” information flow. Since a DNA molecule is a sequence of four elements (A, C, G, and T) and protein is a sequence of twenty types of amino acids, the central dogma of molecular biology implies that biological systems promote a jump in *information content* [39] while synthesizing proteins using genetic information written in DNA. (According to Shannon’s information theory [39], the maximum possible information content of DNA is log_2_(4) = 2 Bits, and the maximum possible information content of protein is log_2_(20) = 4.322 Bits. The maximum possible information content, called decision content, occurs when all constituent elements exhibit the same probability.)

Authors consider that the information processing underlying DNA–RNA–Protein (i.e., the central dogma of molecular biology) and the subsequent jump in information content can be of great use while solving smart manufacturing-relevant problems (pattern recognition and similarity indexing). In this article, the authors use some arbitrary examples and present the operations used in DNA–RNA–Protein-like information processing for solving the abovementioned cognitive problems.

The rest of this article is organized as follows. Section 2 uses a schematic diagram to describe the central dogma of molecular biology and the main processes involved. Section 3 presents algorithms underlying DNA-Based Computing (DBC) inspired by the central dogma of molecular biology. Section 4 presents three arbitrary examples showing the cognitive computing ability of DBC. In particular, the following three types of problems are considered: (1) similarity indexing between seemingly different but inherently identical objects; (2) recognizing regions of an image separated by a complex boundary; and (3) recognizing patterns using insufficient information (e.g., a very short-windowed sensor signal). Section 5 concludes this article.

## 2. Central Dogma of Molecular Biology

As mentioned before, the central dogma of molecular biology establishes the logical and physical relationships among macro molecules such as DNA, RNA, and proteins. In particular, biological systems only allow “DAN to DNA”, “DNA to RNA”, and “DNA to protein” information flow [36,37]. A comprehensive description of DNA–RNA–Protein-centric processes can be found in [38]. In this article, the objective is to gain inspiration from the core processes of central dogma and build models (algorithms) to solve cognitive problems. Therefore, a customized and brief description of the core processes underlying the central dogma of molecular biology is presented in this section.

Figure 1 schematically illustrates the core processes of the central dogma of molecular biology [30,31]. As seen in Figure 1, the three core processes are (1) DNA replication, (2) DNA transcription, and (3) RNA translation. The first one produces a copy of DNA when cell division occurs [38]. The other two are involved in protein synthesis [38]. The enzymatic activities of the latter two processes are denoted as *ENZm* and *ENZt* [28]. The molecules involved are DNA (double strands of nucleic acids (A, C, G, and T), *m*RNA (single strand of nucleic acids (A, C, G, and U)), *t*RNA (a relatively short stand of nucleic acids), codon (three consecutive nucleic acids of *m*RNA), anti-codon (counterpart of codon in *t*RNA), free bases (free nucleic acids), *t*RNA charged with amino acid, and protein (sequence of amino acids). Thus, the main objective of these molecules and enzymatic activities is to synthesize a protein based on the genetic information stored in DNA. Consequently, the central dogma’s main purpose is to maintain a unidirectional information flow (DNA to a protein via RNA but not protein to the other way).

DNA transcription’s enzymatic activities (*ENZm*) first recognize a DNA segment as a gene and make a copy of it (gene), synthesizing free bases. The copied segment is used to form a messenger RNA (*m*RNA). While doing so, A is copied as U, C is copied as G, G is copied as C, and T is copied as A. On the other hand, the enzymatic activities of RNA translation (*ENZt*) operate on *m*RNA, *t*RNA, and free amino acids and produce a protein (sequence of amino acids). In doing so, *ENZt* first bonds an appropriate amino acid to a *t*RNA. The selection of amino acids depends on the anti-codon (a three-base sequence of nucleic acids at a certain location of *t*RNA). In the next step, *ENZt* helps to make a bond between a codon of mRNA and an anti-codon of *t*RNA-amino-acid compound. Finally, *ENZt* helps the *t*RNA-amino-acid compound release the amino acid to the growing chain of the protein. This way, the genetic instruction stored in DNA is passed to proteins. It is worth mentioning that the sequence of amino acids of a protein is its primary structure. It makes a three-dimensional structure due to folding (tertiary structure). The tertiary structures of proteins perform functional activities and serve as structural units in biological systems. Therefore, biological systems keep producing proteins. Other than DNA transcription and RNA translation, there is an important process that is called DNA replication by which the DNA is replicated for storage and cell division. Some microorganisms rely on *m*RNA directly for protein synthesis.

However, there are some universal rules that biological systems use during DNA transcription and RNA translation. These are called genetic rules. The genetic rules establish relationships among amino acids, codons of *m*RNA, and anti-codons of *t*RNA. Table 1 shows the nucleic acid relations of DNA, *m*RNA, and *t*RNA.

## 3. DNA-Based Computing (DBC)

Computing methods can be developed inspired by the core processes involved in the central dogma of molecular biology. The authors denote these computing methods as DNA-Based Computing (DBC) [30,31,32,33,34,35]. DBC can take many forms, but the basic principle remains the same: generate many-element-based pieces of information (similar to protein) from a few-element-based pieces of information (similar to DNA and RNA). Consequently, a rise in the information content [39] may take place (see Section 1). The two most promising forms (algorithms) of DBC are presented below. The first is denoted as DBC-1 and the other as DBC-2. These algorithms are described as follows.

First, consider DBC-1. It is schematically illustrated in Figure 2. This algorithm consists of four steps and seven processes. Referring to Figure 2, the steps and processes are described as follows.

*Step 1*: This step consists of Process 1. Process 1 extracts problem-relevant information from a given problem.

*Step 2*: This step consists of Processes 2 and 3 and generates DNA-like information (DNA array) from the problem-relevant information. In doing so, the user sets the DNA forming rules. For example, consider that a binary string <0000111001110000011111111> is the problem-relevant information. If the user defines that 00 → A, 01 → C, 10 → G, and 11 → T, which are denoted as DNA translation rules, then the resulting DNA array is <AATGCTAACTTT>. Note that the last digit (“1”) is truncated. Instead of truncating a digit, one more digit can be added (i.e., in this case “1” or “0” depending on a user-defined process). In that case, the resulting DNA array becomes one letter longer, i.e., <AATGCTAACTTTT>. The reading frame is also a constituent of DNA-forming rules. For example, the above case refers to the pair-wise reading frame, i.e., two consecutive letters in the binary string are replaced by a letter of DNA. There are other possibilities. Each digit in the binary string can be replaced by a letter of DNA, considering it (the digit) and the next one. This reading frame is denoted as the continuous reading frame. Thus, the continuous reading frame converts <0000111001110000011111111> to <AAACTTGACTTGAAAACTTTTTTTT>. Therefore, the constituents of DNA-forming rules are DNA translation rules, truncation/adding schemes, and reading frames.

*Step 3*: This step consists of Processes 4 and 5 and produces a protein array that consists of single-letter symbols of amino acids. In doing so, protein-forming rules use protein translation rules as shown in Table 2 (i.e., three consecutive letters of DNA array are considered a codon and translated into the corresponding single-letter symbol of protein). It also employs a truncation/addition scheme and a reading frame similar to the ones described in the previous process. For example, if the DNA array <AATGCTAACTTTT> undergoes the protein-forming rules, the following protein array can be produced: <*NMCALXNTLF*>. In this case, the protein-forming rule is DNA(*i*)DNA(*i* + 1)DNA(*i* + 2) = codon(*k*) → AA(*k*), *i* = 1,2,…, *k* = 1,2,…, where DNA(*i*) is the *i*-th letter of the DNA array, codon(*k*) is the *k*-th codon (one of the three-letter DNA bases shown in Table 2), and AA(*k*) is the corresponding single letter symbol of the amino acid (according to Table 2). Thus, ∀codon(*k*) ∈ {AAA, AAC, AAG, AAT, ACA, ACC, ACG, ACT, AGA, AGC, AGG, AGT, ATA, ATC, ATG, ATT, CAA, CAC, CAG, CAT, CCA, CCC, CCG, CCT, CGA, CGC, CGG, CGT, CTA, CTC, CTG, CTT, GAA, GAC, GAG, GAT, GCA, GCC, GCG, GCT, GGA, GGC, GGG, GGT, GTA, GTC, GTG, GTT, TAA, TAC, TAG, TAT, TCA, TCC, TCG, TCT, TGA, TGC, TGG, TGT, TTA, TTC, TTG, TTT}. Consequently, a protein array denoted as *protein* = <…AA(*k*)…> is the outcome of the translation process codon(*k*) = DNA(*i*)DNA(*i* + 1)DNA(*i* + 2) → AA(*k*) governed by the following protein-forming rules.

IF codon(*k*) ∈ {ATT, ATC, ATA} THEN protein(*k*) = *I*IF codon(*k*) ∈ {CTT, CTC, CTA, CTG, TTA, TTG} THEN protein(*k*) = *L*IF codon(*k*) ∈ {GTT, GTC, GTA, GTG} THEN protein(*k*) = *V*IF codon(*k*) ∈ {TTT, TTC} THEN protein(*k*) = *F*IF codon(*k*) ∈ {ATG} THEN protein(*k*) = *M*IF codon(*k*) ∈ {TGT, TGC} THEN protein(*k*) = *C*IF codon(*k*) ∈ {GCT, GCC, GCA, GCG} THEN protein(*k*) = *A*IF codon(*k*) ∈ {GGT, GGC, GGA, GGG} THEN protein(*k*) = *G*IF codon(*k*) ∈ {CCT, CCC, CCA, CCG} THEN protein(*k*) = *P*IF codon(*k*) ∈ {ACT, ACC, ACA, ACG} THEN protein(*k*) = *T*IF codon(*k*) ∈ {TCT, TCC, TCA, TCG, AGT, AGC} THEN protein(*k*) = *S*IF codon(*k*) ∈ {TAT, TAC} THEN protein(*k*) = *Y*IF codon(*k*) ∈ {TGG} THEN protein(*k*) = *W*IF codon(*k*) ∈ {CAA, CAG} THEN protein(*k*) = *Q*IF codon(*k*) ∈ {AAT, AAC} THEN protein(*k*) = *N*IF codon(*k*) ∈ {CAT, CAC} THEN protein(*k*) = *H*IF codon(*k*) ∈ {GAA, GAG} THEN protein(*k*) = *E*IF codon(*k*) ∈ {GAT, GAC} THEN protein(*k*) = *D*IF codon(*k*) ∈ {AAA, AAG} THEN protein(*k*) = *K*IF codon(*k*) ∈ {CGT, CGC, CGA, CGG, AGA, AGG} THEN protein(*k*) = *R*IF codon(*k*) ∈ {TAA, TAG, TGA} THEN protein(*k*) = *X*

*Step 4*: This is the last step. It consists of Processes 6 and 7 and generates a solution to solve the problem. In this case, the information of the protein array (both sequential information and frequency-driven information) is used to extract problem-solving rules. Here, sequential information means whether a particular amino acid sequence occurs. Frequency-driven information means which amino acids are predominant, which are not, and which are absent in the protein array. See Section 4 for examples of problem-solving rules.

The other form of DBC, DBC-2, is schematically illustrated in Figure 3.

Compared to DBC-1, DBC-2 has an additional functionality to deal with *m*RNA-forming rules and *m*RNA array. In addition, it produces multiple DNA arrays unlike DBC-2. Consequently, DBC-2 consists of five steps and nine processes. The description is as follows.

*Step 1:* This step consists of Process 1 and produces problem-relevant information from a given problem. Thus, this step is similar to that of Step 1 of DBC-1.

*Step 2:* This step consists of Processes 2 and 3 and generates multiple DNA arrays using different DNA-forming rules. Recall the arbitrary example of the DNA array of DBC-1 where <0000111001110000011111111> is converted to <AATGCTAACTTT> using the following DNA-forming rules: DNA translation rules: 00 → A, 01 → C, 10 → G, and 11 → T; truncation/addition scheme: truncation; and pair-wise reading frame. If a different DNA translation is used, i.e., 11 → A, 01 → C, 10 → G, and 00 → T, keeping truncation/addition and reading frame the same, then the following DNA array results <TTAGCATTCAAA>. Similarly, if another different DNA translation is used, i.e., 11 → A, 10 → C, 01 → G, and 00 → T, keeping truncation/addition and reading frame the same, then the following DNA array results <TTACGATTGAAA>. This way, multiple DNA arrays can be produced from a single piece of problem-relevant information.

*Step 3:* This step consists of Processes 4 and 5 and generates an *m*RNA array using *m*RNA-forming rules. The *m*RNA-forming rules determine how to integrate the DNA arrays produced in the previous step while generating an *m*RNA array. For the sake of better understanding, three types of *m*RNA-forming rules are presented as follows. These rules are denoted as replication rule, direct addition rule, and cascading rule. The resulting *m*RNA arrays are denoted as replicated *m*RNA, added *m*RNA, and cascaded *m*RNA, respectively. The replication rule is described as follows. Let DNA array be DNA = <AATGCTAACTTT> where DNA(*i*), *i* = 1, 2, …, denotes an element of the DNA array, e.g., DNA(*i* = 5) = C. A replication rule produces a replicated *m*RNA as follows: *m*RNA = <…DNA(*i*)DNA(*i*)…> = <AAAATTGGCCTTAAAACCTTTTTT>. In this particular replication rule, the same DNA is used twice (i.e., DNA is replicated) to create a replicated *m*RNA. The same DNA can be replicated as many times as preferred while producing a replicated *m*RNA. On the other hand, a direct addition rule operates on multiple but different DNA arrays. These DNA arrays are produced from the same piece of problem relevant information. This means that a set of DNA-forming rules is used to produce multiple DNA arrays from the same piece of problem relevant information. For example, consider the following formulation. Let the DNA arrays be DNA1 = <AATGCTAACTTT>, DNA2 = <TTAGCATTCAAA>, and DNA3 = <*TTACGATTGAAA*>. A direct addition rule produces the following added *m*RNA: *m*RNA = <DNA1DNA2DNA3>, i.e., *m*RNA = < AATGCTAACTTTTTAGCATTCAAA*TTACGATTGAAA*>. This means that the DNA arrays are simply appended one after another to produce an added *m*RNA. A direct addition rule can handle as many DNA arrays as preferred, i.e., there is no restriction on the number of DNA arrays. Finally, a typical cascading rule is described as follows. A cascading rule also needs multiple but different DNA arrays. It appends the DNA arrays elementwise, not one after another. For example, recall the DNA arrays used in the previous case (direct addition rule), i.e., DNA1, DNA2, and DNA3. The respective elements of the DNA arrays are denoted as DNA1(*i*), DNA2(*i*), and DNA3(*i*), respectively. Thus, cascading means appending the elements DNA1(*i*), DNA2(*i*), and DNA3(*i*), respectively. This results in a cascading *m*RNA as follows: *m*RNA = <…*m*RNA(*j*)*m*RNA(*j* + 1)*m*RNA(*j* + 2)…> = <…DNA1(*i*)DNA2(*i*)DNA3(*i*)…> = <A**T***T*A**T***T*T**A***A*G**G***C*C**C***G*T**A***A*A**T***T*A**T***T*C**C***G*T**A***A*T**A***A*T**A***A*>. This means that three consecutive elements of mRNA, *m*RNA(*j*), *m*RNA(*j* + 1), *m*RNA(*j* + 2), *j* = 1,2,…, are taken from the respective elements of DNA arrays, i.e., *m*RNA(*j*) = DNA1(*i*), *m*RNA(*j* + 1) = DNA2(*i*), and *m*RNA(*j* + 2) = DNA3(*i*).

*Step 4:* This step consists of Processes 6 and 7 and produces a protein array that consists of single-letter symbols of amino acids. This step is similar to Step 3 of DBC-1. The only difference is it operates on *m*RNA array not on the DNA array(s). Thus, this step is similar to that of Step 1 of DBC-1.

*Step 5:* This is the last step. It consists of Processes 8 and 9 and generates a solution to solve the problem. It is similar to Step 4 of DBC-1.

Other than DBC-1 and DBC-2, other forms of DBC can be developed based on the basic principle: obtain a many-element piece of information (similar to protein) from a few-element piece of information (similar to DNA). One of the remarkable things is that a user has much freedom to customize DBC for a given problem. This freedom is exercised by fixing the DAN-forming, *m*RNA-forming, protein-forming, and problem-solving rules. Protein-forming rules are fixed unless the user wants to replace the genetic rules (Table 2) with other rules that obey the abovementioned basic principle. Regarding problem-solving rules, DBC must not act like a black-box type machine learning algorithm (e.g., artificial neural network). The focus is on engaging human users with the problems they want to solve. The case studies presented in the next section shed light on the performance and characteristics of DBC described above.

## 4. Results

This section presents three arbitrary illustrative examples showing the cognitive computing ability of DBC-1 and DBC-2. The first example deals with similarity indexing between seemingly different but inherently the same objects. The second example deals with recognizing two regions of an image separated by a complex boundary. The last example deals with pattern recognition when the relevant information is insufficient.

### 4.1. Similarity Indexing

In many smart manufacturing applications, it is necessary to exchange decision-relevant information among independent workspaces. In this case, similarity indexing between seemingly different but inherently the same objects can help to build trust in the information exchanged [31]. Ullah et al. [31] and D’Addona et al. [32] used DBC to index seemingly different but inherently identical images. This subsection presents the key aspects of the DBC-based indexing procedure. For the sake of better understudying, a well-knowledge shape (tree) modeled by fractal geometry [40] is considered.

Figure 4 shows four samples of the tree. A set of 1000 points is used to represent each tree. The mathematical settings for generating the points can be found in [41]. Since a stochastic process is involved, the positions of the points are not the same for the trees, but they collectively represent the same object (a tree). This similarity of these trees can be verified using DBC-1. The description is as follows.

The problem here is to find out the similarity of the trees (Figure 4) in a quantitative manner using DBC-1. Table 3 summarizes the settings of DBC-1 used in this case.

The frequencies (*fr*(.)) of amino acids in the proteins of the trees in Figure 4a–d (except *K*) are plots as shown in Figure 5. It is found that the frequencies of *K* are the same (8016) for the trees in Figure 4a–c, whereas it is 8293 for the tree in Figure 4d. Since the frequency of *K* is very large compared to those of others for each tree, the frequencies of other amino acids (except *K*) are plotted, as shown in Figure 5, to understand the variability in the frequencies of amino acids. These plots also help extract rules for solving the problem (similarity indexing). In synopsis, the following rules are extracted using the results shown in Figure 5. The extracted rules are summarized in Table 4 and described as follows. The zero-frequency amino acids are *G*, *H*, *I*, *M*, *P*, *Q*, *S*, *V*, *W*, and *Y*. The equal frequency amino acids are *E* and *N*. In particular, *fr*(*E*) = *fr*(*N*) = 341 for the trees (a)–(c) in Figure 5 and *fr*(*E*) = *fr*(*N*) = 273 for the tree in Figure 4d. In addition, the summation of frequencies of *D* and *N* is equal to the frequency of *T*, i.e., (*fr*(*D*) + *fr*(*N*) = *fr*(*T*) = 383 for trees (a)–(c) in Figure 5 and 330 for the tree (d) in Figure 5. The least frequent amino acid other than the zero-frequency ones is *C*. Finally, the list of amino acids in ascending order of frequencies is as follows: *fr*(*A*) < fr(*X)* < *fr*(*L)* < *fr*(*R*) < *fr*(*E*) < *fr*(*N*) < *fr*(*T*). Therefore, observing whether or not the rules shown in Table 4 are obeyed, it can be confirmed that the shapes (this time the four trees in Figure 5) are the same. This way, a set of transparent and human-comprehensible rules can be extracted from the sequential and frequential states of the amino acids in the proteins. These rules can be used to decide whether or not the objects are similar or not.

### 4.2. Image Processing

In some applications of smart manufacturing, image processing is needed. For example, Kubo et al. [35] performed a topographical analysis of an image of a grinding wheel using DBC to see the effectiveness of dressing operations that sharpen the wheel. Iwadate and Ullah [33] determined the outer boundary of a complex shape given by a point cloud using DBC. In such applications, stochastically distributed segments of an image must be recognized and processed to obtain the desired result. This can be achieved easily if different amino acids could have been assigned the stochastically distributed segments. In doing so, some specific setting of DBC is needed. This subsection describes the specific setting using an arbitrary example as follows.

Consider an arbitrary image, as shown in Figure 6, where similar objects are inside and outside a complex boundary. How to eliminate the objects within the boundary while keeping those outsides is the problem here. In other words, the problem is determining a convex or concave hull representing the boundary and doing the rest, i.e., eliminating similar objects inside the hull while keeping those outside.

To be more specific, consider a particular case (the shape of a snowflake), as shown in Figure 7. As seen in Figure 7, the snowflake is represented by a (100 × 100) two-dimensional binary array. There are 0 s inside and outside of the boundary of the snowflake, whereas 1 s reside inside the snowflake only. If 0 s inside the snowflake convert to 1 s and 0 s outside of the snowflake remain unchanged, then 1 s represent the snowflake itself, and 0 s represent the outside of the snowflake. Thus, the above operation results in a concave hull (boundary) that defines the geometric shape of the snowflake. In order to perform the above-mentioned concave hulling, the settings of DBC-1 shown in Table 3 can be used. This is what is performed, and the results are shown in Figure 7, Figure 8, Figure 9, Figure 10 and Figure 11. Figure 8 shows the two-dimensional DNA array of the snowflake and its surroundings. An element of DNA array is denoted as DNA(*p*,*q*) ∈ {A, C, G, T}, *p* = 1,2,…, *q* = 1,2,…. Figure 9 shows the codons generated from the two-dimensional DNA array (Figure 7); three consecutive elements of a row of the DNA array are used to create a codon, i.e., codon(*j*,*k*) = DNA(*p*,*q*)DNA(*p* + 1,*q*)DNA(*p* + 2,*q*), for all *p* = 1,2,…, *q* = 1,2,…, *j* = 1,2,…, and *k* = 1,2,…. Figure 10 shows the two-dimensional protein array (protein = <…protein(*j*,*k*) = AA(*j*,*k*)…>) generated from the codon array (Figure 9). In this case, the protein-forming rule converts a codon to an amino acid, i.e., codon(*j*,*k*) → AA(*j*,*k*), *j* = 1,2,…, *k* = 1,2,…, according to the genetic rules shown in Table 2. Thus, ∀codon(*j*,*k*) ∈ {AAA, AAC, AAG, AAT, ACA, ACC, ACG, ACT, AGA, AGC, AGG, AGT, ATA, ATC, ATG, ATT, CAA, CAC, CAG, CAT, CCA, CCC, CCG, CCT, CGA, CGC, CGG, CGT, CTA, CTC, CTG, CTT, GAA, GAC, GAG, GAT, GCA, GCC, GCG, GCT, GGA, GGC, GGG, GGT, GTA, GTC, GTG, GTT, TAA, TAC, TAG, TAT, TCA, TCC, TCG, TCT, TGA, TGC, TGG, TGT, TTA, TTC, TTG, TTT}. Consequently, a protein array denoted as *protein* = <…AA(*j*,*k*)…> is the outcome of the translation process codon(*j*,*k*) = DNA(*p*,*q*)DNA(*p* + 1,*q*)DNA(*p* + 2,*q*) → AA(*j*,*k*) governed by the following protein-forming rules. As seen in Figure 10, the outside of the snowflake is occupied by the amino acid denoted as *K*. The inside of the snowflake is occupied by amino acids other than *K*, but, still, there are exceptions. These few *K*s inside the boundary of the snowflake can be converted to an amino acid other than *K* using a simple rule, as follows:(1)$$\left(\left(protein\left(j,k-1\right)=\neg K\right)\wedge \left(protein\left(j,k\right)=K\right)\wedge \left(protein\left(j,k+1\right)=\neg K\right)\to c\_protein\left(j,k\right)=\left(\neg K\right)\right)$$

The above rule takes three consecutive amino acids from a row of the protein array, *protein*(*j*,*k*−1)*protein*(*j*,*k*)*protein*(*j*,*k* + 1), and checks whether the middle amino acid is *K* and the other two are not *K*. If this condition is true, the middle amino acid is replaced by an amino acid other than *K*. Otherwise, the rule does not make any change in the protein array. This way, a corrected protein array (*c_protein*) is created. Thus, the above rule is the problem-solving rule for this particular case. The corrected protein array is shown in Figure 11. As seen in Figure 11, *K*s (pink color) occupy the outside area of the snowflake, and amino acids other than *K* occupy the inside area of the snowflake. The corrected protein can be converted into a binary array by replacing all non-*K*s with 1s and all *K*s with 0s. Consequently, this new binary array becomes the geometric model of the snowflake.

### 4.3. Pattern Recognition in Time Series Datasets

In smart manufacturing, manufacturing enablers come equipped with sensors that collect signals in the form of time series datasets. Machine learning methods like artificial neural networks are often employed to learn from time series datasets. The knowledge learned is then used to make informed decisions, ensuring the sustainable operations of the manufacturing enablers. The remarkable thing is that a large number of datasets are required to learn and use the knowledge due to the inherent characteristics of the abovementioned machine learning methods. Consequently, when the abovementioned machine learning-based methods are involved in a decision cycle, they slow it down. This is a bottleneck because the decision cycle (learning and using required knowledge) must be very fast in smart manufacturing; otherwise, sustainable operation cannot be guaranteed. Thus, a new breed of machine learning methods is needed to support the decision cycles of smart manufacturing adequately. DBC can contribute in this regard. Ullah [30] made a study on this. Based on that study, this subsection presents the salient points of DBC when it acts as a less-data-dependent and quick machine learning method.

From the perspective of DBC, learning the underlying patterns possessed by a given time series dataset is based on one-to-one mapping. Here, one-to-one mapping means mapping a point *x*(*t*) ∈ ℛ of a time series dataset, *TS* = <*x*(*t*) | 0,1,…>, to an amino acid. For this reason, at least three DNA arrays are needed, and these DNA arrays must produce an *mRNA* based on the cascading rule (see Section 3). The DNA arrays can be the same or different ones. As a result, a codon (three consecutive letters of *m*RNA) represents a point (*x*(*t*)). In other words, an amino acid in the protein array (*protein*(*t*)) corresponds to a point, *x*(*t*), of a time series *TS* = <*x*(*t*) | 0,1,…>. Therefore, the authors recommend DBC-2 (not DBC-1) to solve the pattern recognition problems associated with time series datasets.

Unlike the DNA-forming rules described above, new DNA-forming rules are used to recognize patterns from a given time series. The new rules are schematically illustrated in Figure 12. As seen in Figure 12, first, a signal is subtracted from a user-defined baseline denoted as *B* = <…*B*(*t*)…>. This results in a difference time series dataset, denoted as *diff* = <…*d*(*t*)…>. Thus, the following relationship holds: *d*(*t*) = *x*(*t*) – *B*(*t*), ∀*t* = {0,1,…}. The difference time series dataset is processed using four thresholds, *a* > *b* > *c* > *d*, to create a DNA array. For this, the following rules are used: *d*(*t*) ∈ [*c*, *b*] → A, *d*(*t*) ∈ (*b*, *a*) → C, *d*(*t*) ∈ (*d*, *c*) → G, (*d*(*t*) ≥ *a* or *d*(*t*) ≤ *d*)) → T. The arbitrary case shown in Figure 12 results in a DNA array DNA = <AGACATCAATAACAAAAAAAA>. This way, several DNA arrays can be produced. Each time, baseline and thresholds can be set as preferred by the user.

Consider two time series datasets. One follows a normal distribution with a mean of 100 and a standard deviation of 5. It is denoted as a normal signal. The other follows a random distribution in the interval [85, 115]. It is denoted as a non-normal signal. The time series datasets are shown in Figure 13 using a plot. From the visual inspection (Figure 13), it is clear that these two datasets are hardly distinguishable. DBC-2 must make them distinguishable. The formulations shown in Table 5 are used, and protein arrays (this time one-dimensional) are generated to determine whether DBC-2 can distinguish these two signals. The time series datasets are created many times using Monte Carlo simulation, and the corresponding protein arrays are recorded. The first 25 samples of the protein arrays and their analysis results are shown in Table 6. As seen in Table 6, in both cases, the protein arrays consist of the following four amino acids: *F*, *G*, *K*, and *P*. The amino acid “*K*” dominates the normal time series datasets, whereas the amino acid “*F*” dominates the non-normal ones. The Shannon’s entropy function [39] is used to calculate the entropy or average information content of each protein, as follows:(2)$$En\left(protein\left(.\right)\right)=\sum_{i=1}^{4}Pr\left({aa}_{i}\right)\log_{4}\left(\frac{1}{Pr\left({aa}_{i}\right)}\right)\;dna$$

In Equation (2), *En*(*protein*(.)) ∈ [0, 1] denotes the entropy of a given protein where the unit is “dna”, indicating the logarithm function has a base equal to four, *aa*_1_ = *F*, *aa*_2_ = *G*, *aa*_3_ = *K*, *aa*_4_ = *P*, and *Pr*(*aa_i_*) is the probability of *aa_i_* in the given protein array. The probability is calculated by dividing the frequency of *aa_i_* (*fr*(*aa_i_*)) by the number of amino acids in the given protein array. Since each time series dataset consists of 21 amino acids (Table 6), the total number of amino acids is 21. Thus, the following relation holds: *Pr*(*aa_i_*) = *fr*(*aa_i_*)/21.

It is observed that the entropy of a normal signal is less than that of a non-normal one. In rare cases, the entropy of a normal signal becomes greater than or equal to that of a non-normal one. For example, consider the number 3 protein arrays in Table 6. In this case, both protein arrays have the same entropy, but the frequency of *F* is high for the non-normal protein array, and the frequency of *K* is high for the normal protein array. Thus, a logical test regarding the values of entropy of the respective proteins and frequencies of *F* and *K* can be used as a problem-solving rule to detect whether or not a signal is a normal signal, though it may look different. Regarding Table 6, the following rule is applied: *En*(*protein*(normal) > *En*(*protein*(non-normal), or otherwise *fr*(*K*) > *fr*(*F*). When this rule holds, “yes” is returned, as shown in Table 6. In addition, only two of a hundred repetitions produce an incorrect pattern recognition. This means that DBC-2 is an effective machine learning algorithm that recognizes patterns in time series datasets even though the time series window is very short.

## 5. Conclusions

Employing bio-inspired computing has been proven to be one of the effective ways to develop intelligent systems for smart manufacturing. DBC, based on the central dogma of molecular biology, is a valuable constituent of bio-inspired computing. However, it is less popular than other bio-inspired computing methods, such as artificial neural networks and genetic algorithms/programming. Especially when the transparency of a machine learning algorithm is an issue, DBC is a better choice. Using simple mapping from few-element to many-element, DBC provides a simple solution to complex cognitive problems. The human intervention can also be captured very easily because DNA- and mRNA-forming rules are user-defined, and these rules explicitly capture the human intervention in the whole computing process. Having said that, the authors do not mean that DBC is a competitor of other bio-inspired computing methods found in the literature. Rather, the authors mean that it is a synergetic method that deserves further exploration.

The problems addressed in this article—(1) similarity indexing between seemingly different but inherently the same objects, (2) recognizing two regions of an image separated by a complex boundary, and (3) pattern recognition when the relevant information is insufficient—are common cognitive problems in smart manufacturing. Thus, DBC can be positioned in a smart manufacturing framework, as schematically illustrated in Figure 14. As seen in Figure 14, DBC occupies the cyberspace of smart manufacturing. The cognitive problems (CPs) associated with the ongoing manufacturing activities of the Industrial Internet of Things-based enablers can be integrated with DBC. DBC acknowledges the cognitive problems and consults documents and datasets collected from ongoing and past manufacturing activities. Finally, the solutions to CPs generated by DBC are fed into the manufacturing enablers.

## Figures and Tables

**Figure 1 biomimetics-08-00620-f001:**
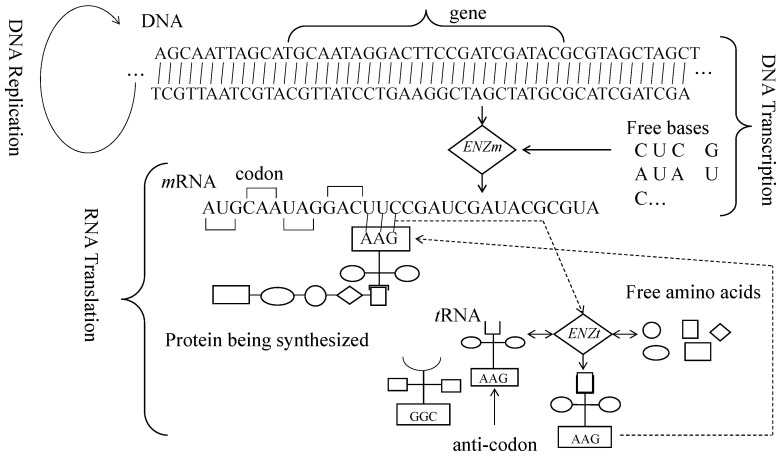
Core processes underlying the central dogma of molecular biology [28,30,31].

**Figure 2 biomimetics-08-00620-f002:**
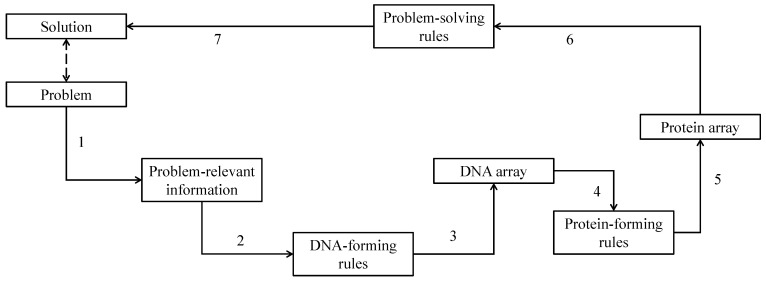
Flowchart of DBC-1.

**Figure 3 biomimetics-08-00620-f003:**
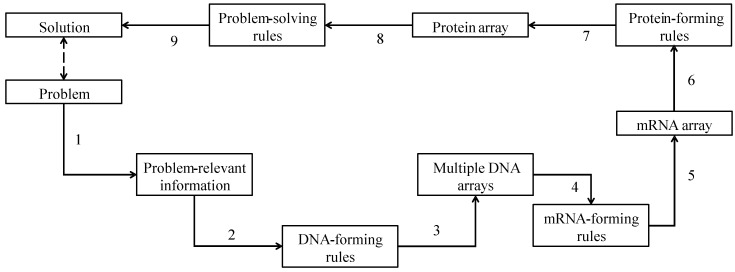
Flowchart of DBC-2.

**Figure 4 biomimetics-08-00620-f004:**
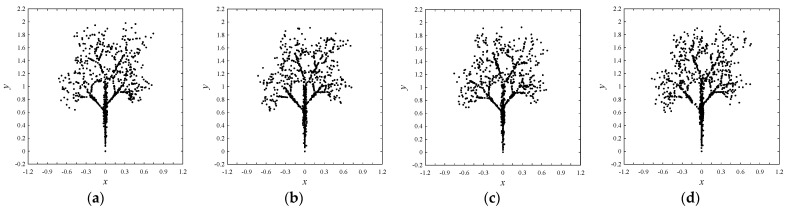
Four trees each represented by 1000 points. (**a**–**d**) Four trees wherein each tree is modeled by 1000 points generated stochastically using fractal geometry.

**Figure 5 biomimetics-08-00620-f005:**
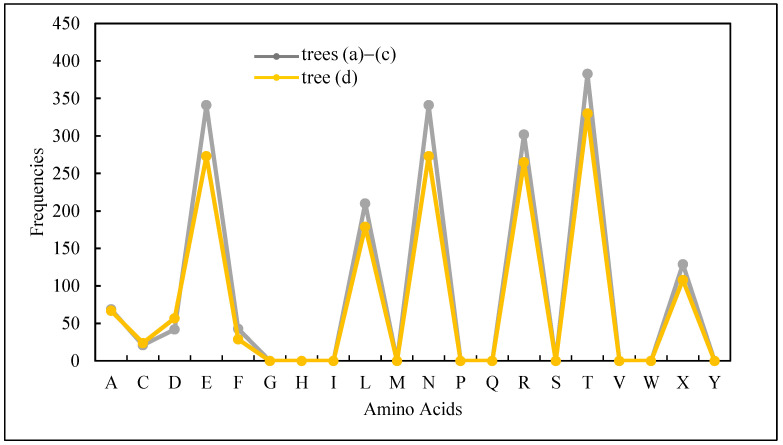
Frequencies of amino acids (except *K*) in the protein arrays of the four trees.

**Figure 6 biomimetics-08-00620-f006:**
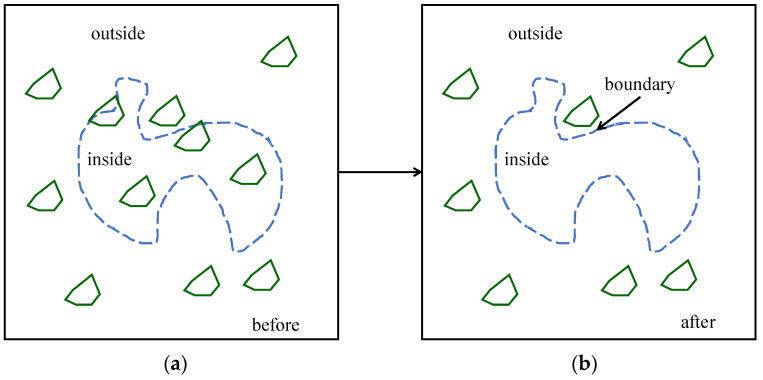
A geometric modeling problem. (**a**) Before geometric modeling; (**b**) after geometric modeling.

**Figure 7 biomimetics-08-00620-f007:**
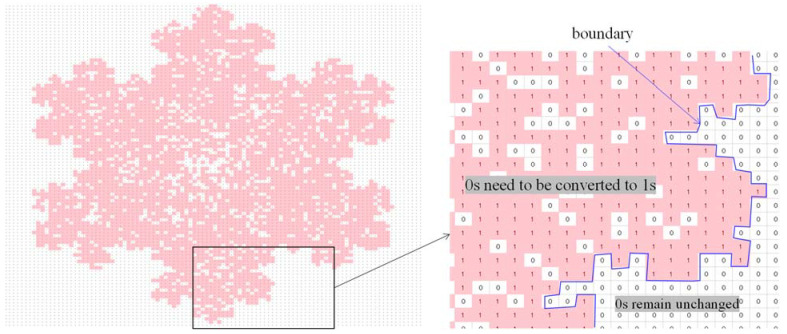
A snowflake in the form of a two-dimensional binary array.

**Figure 8 biomimetics-08-00620-f008:**
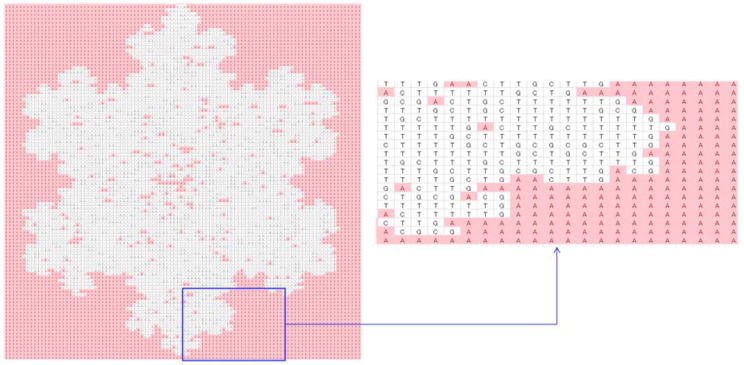
Two-dimensional DNA array of the snowflake.

**Figure 9 biomimetics-08-00620-f009:**
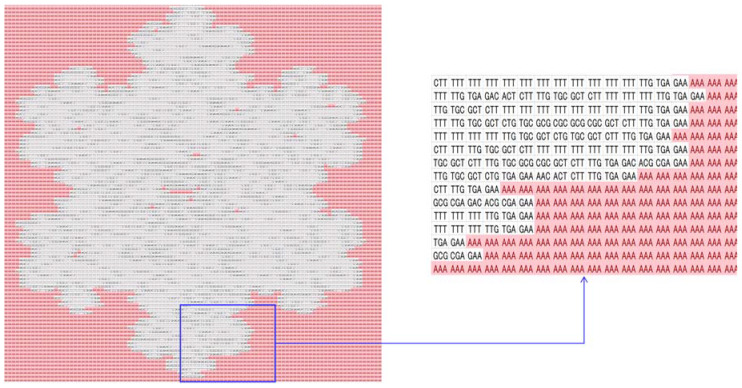
Codons generated from the two-dimensional DNA array (Figure 7).

**Figure 10 biomimetics-08-00620-f010:**
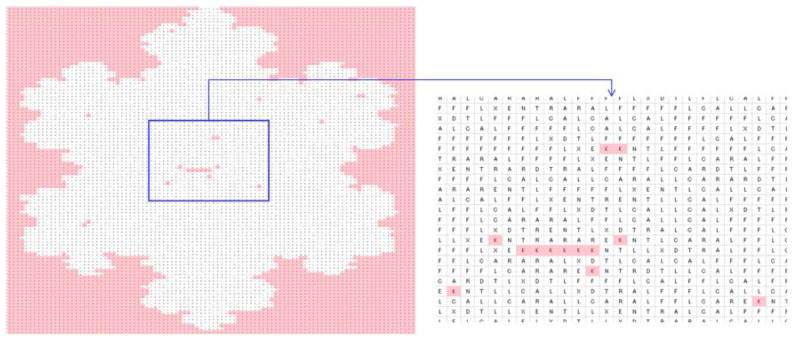
Two-dimensional protein array generated from the codon array (Figure 8).

**Figure 11 biomimetics-08-00620-f011:**
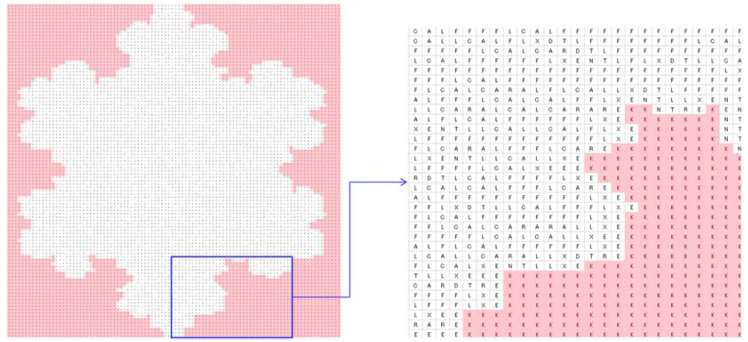
Two-dimensional corrected protein array generated from the protein array in Figure 9.

**Figure 12 biomimetics-08-00620-f012:**
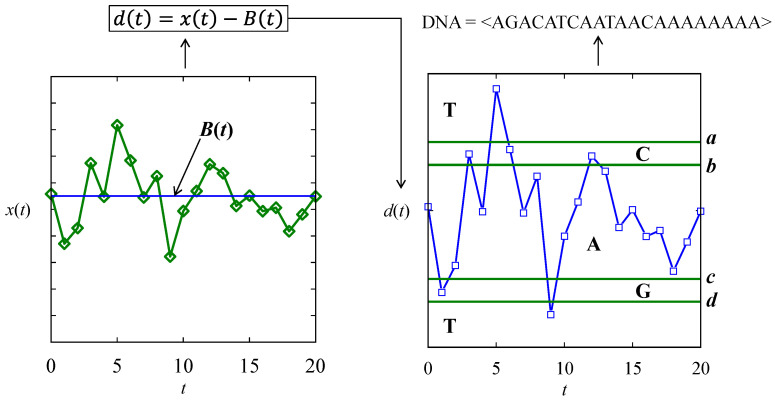
DNA-forming rules for time series datasets.

**Figure 13 biomimetics-08-00620-f013:**
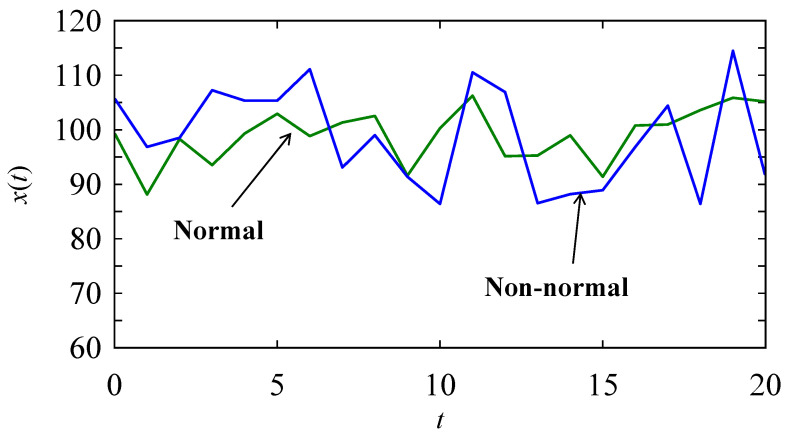
Two stochastic time series datasets.

**Figure 14 biomimetics-08-00620-f014:**
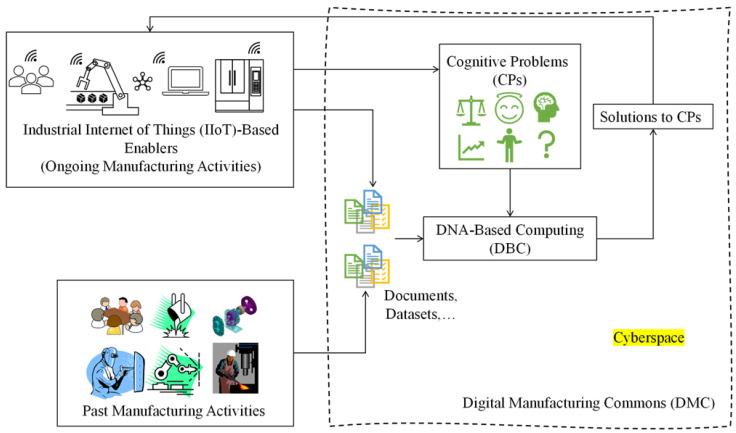
Perceived role of DBC in smart manufacturing.

**Table 1 biomimetics-08-00620-t001:** Nucleic acids relationships [30,31,38].

DNA	*m*RNA (Codon)	*t*RNA (Anti-Codon)
A	U	A
C	G	C
G	C	G
T	A	U
Nucleic acid symbols		

**Table 2 biomimetics-08-00620-t002:** Amino acid and nucleic acid relationships [30,31,38].

No	Three-Letter DNA Bases (Codons ^$^)	Amino Acids (Single-Letter Symbols)
1	ATT, ATC, ATA	Isoleucine (*I*)
2	CTT, CTC, CTA, CTG, TTA, TTG	Leucine (*L*)
3	GTT, GTC, GTA, GTG	Valine (*V*)
4	TTT, TTC	Phenylalanine (*F*)
5	ATG	Methionine (*M*)
6	TGT, TGC	Cysteine (*C*)
7	GCT, GCC, GCA, GCG	Alanine (*A*)
8	GGT, GGC, GGA, GGG	Glycine (*G*)
9	CCT, CCC, CCA, CCG	Proline (*P*)
10	ACT, ACC, ACA, ACG	Threonine (*T*)
11	TCT, TCC, TCA, TCG, AGT, AGC	Serine (*S*)
12	TAT, TAC	Tyrosine (*Y*)
13	TGG	Tryptophan (*W*)
14	CAA, CAG	Glutamine (*Q*)
15	AAT, AAC	Asparagine (*N*)
16	CAT, CAC	Histidine (*H*)
17	GAA, GAG	Glutamic acid (*E*)
18	GAT, GAC	Aspartic acid (*D*)
19	AAA, AAG	Lysine (*K*)
20	CGT, CGC, CGA, CGG, AGA, AGG	Arginine (*R*)
21	TAA, TAG, TGA	None (*X* ^%^)

^$^ For computational purposes, the three-letter DNA bases are considered codons. ^%^ The amino acid denoted as *X* does not exist, i.e., TAA, TAG, and TGA do not code any amino acids. For computational purposes, *X* is used to replace TAA, TAG, and TGA.

**Table 3 biomimetics-08-00620-t003:** Settings of DBC-1 for analyzing the trees in Figure 4.

Items	Settings
Problem relevant information	A (100 × 100) binary array of the tree
DNA-forming rules	00 = A, 01 = C, 10 = G, 11 = T, continuous reading frame; truncation/addition: truncation
Protein-forming rules	As described in Section 3

**Table 4 biomimetics-08-00620-t004:** Rules for similarity index of the trees.

Rules	
Zero-frequency amino acids	*G*, *H*, *I*, *M*, *P*, *Q*, *S*, *V*, *W*, and *Y*
Equal-frequency amino acids	*E* and *N* (341 for trees Figure 4a–c and 273 for Figure 4d)
Special frequencies	Summation of frequencies of *D* and *N* is equal to the frequency of *T*, i.e., (*fr*(*D*) + *fr*(*N*) = *fr*(*T*) = 383 for trees Figure 4a–c and 330 for the tree in Figure 4d)
The least frequent amino acid other than the zero-frequency ones	*C*
List of amino acids in the ascending order of frequencies	*fr*(*A*) < fr(*X)* < *fr*(*L)* < *fr*(*R*) < *fr*(*E*) < *fr*(*N*) < *fr*(*T*)

**Table 5 biomimetics-08-00620-t005:** Settings of DBC-2 for pattern recognition of time series datasets.

Items	Settings
Baseline and thresholds	*B*(*t*) = 100, *a* = 10, *b* = 5, *c* = −5, and *d* = 10 for all three DNA arrays
mRNA-forming rules	Cascading rule among the three DNA: DNA1,…,DNA3 *m*RNA = <…codon(*i*) = DNA1(*i*)DNA2(*i*)DNA3(*i*)…>
Protein-forming rules	As described in Section 3

**Table 6 biomimetics-08-00620-t006:** Samples of proteins and their analysis.

No	Protein Arrays (Normal)	Proteins Arrays (Not Normal)	Entropy [dna] (Normal)	Entropy [dna] (Non-Normal)	Problem-Solving Rule Holds
1	*KKKPFKKKKKPKKGKKPKFKK*	*KFKGPFKFKKGFKKGPFGFKK*	0.640	0.910	yes
2	*KGKKKKKKKKKKKKKPKKKGP*	*FKKGFPGGGKKPKFKFPKGKK*	0.446	0.937	yes
3	*PKGKKGKPPPPGKFKKKKKKK*	*PGKKKFPFFFFFKFKFFFFFP*	0.782	0.782	yes
4	*PKKPGKPKFFFKKKKPPKKKP*	*FFGKFKGKKKGFKFGPFKGPK*	0.808	0.931	yes
5	*KGKPKPKPKKKKKGKPKKKGK*	*KPFGKPKFFPFFKGFGGPPKF*	0.623	0.985	yes
6	*PKGGKFGKKKKKKKKGKKKKK*	*PPFKKGKGFFFKFFFGGGKKF*	0.610	0.931	yes
7	*KKKGKKKKKKKPKGKKKKKKP*	*FKKGKFKKPFKGFKFKGFKFF*	0.446	0.832	yes
8	*KKKKPKFPKFKKKKKKKKKKK*	*GPKFFKKFFPKGPFGKFGPKP*	0.446	0.991	yes
9	*GKKKKFPPKKKPFKPKGPPPK*	*FPKGKKFFFKKFPGGKGPGKF*	0.842	0.969	yes
10	*PFKKGKKPKKKFKKPKKKKKK*	*GKFGFKKFKFPFFPFGGFKKK*	0.640	0.919	yes
11	*KKFKKKKPKKKKKPKKGKKKK*	*FPFFKGFGKFGGFFGKGKPGK*	0.494	0.936	yes
12	*KKKFKKPKKKFFKPKKKKKKK*	*KFGKFGPKFFGKKPGFKGGGK*	0.512	0.936	yes
13	*KGKKKKKPKKKKKKKKKKKKK*	*FFKPGFGPGKGKGPKFPPFFF*	0.274	0.985	yes
14	*KKKKKPPKKKKKKGGKGGKPP*	*GGKFKPFKKFGKKKFKFFFKK*	0.670	0.824	yes
15	*KKKKKPGKKKKKKKKKPKKPK*	*FKKFGFFKFFFGGGGFGGGPF*	0.428	0.832	yes
16	*PGPKPKKPPKPKKKKKKKKKK*	*PKGGPPPFFFKFFPFFGPKPK*	0.558	0.957	yes
17	*KGKGKPGKKFKKKKPKKKKKP*	*FFFFKKPKGKKFKKKPPFFKK*	0.701	0.824	yes
18	*KGKGKKGKKKKFGKPKPKGKK*	*PKPPFKKKPKFKKFKKFFFPF*	0.727	0.773	yes
19	*GKPKKKKKPFKPPKPKKKKKG*	*FGPFFKFGFFKKKGKGKGFGP*	0.727	0.942	yes
20	*KPKGKGKKKGKKGKGKPKKGK*	*GPFGKGKGGFFFFGPGPPGKP*	0.634	0.959	yes

## Data Availability

The data are available upon request to the corresponding authors.

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
