# Peer review of "Biologicalization of Smart Manufacturing Using DNA-Based Computing"

_biomimetics, 2023, doi:10.3390/biomimetics8080620_

Round 1

Reviewer 1 Report

Comments and Suggestions for Authors

-the production of mRNA  needs to be explained further  p7,line267

-figure 8 needs process explanation using codons p12

Author Response

our answers are attached. thank you.

Reviewer 2 Report

Comments and Suggestions for Authors

This paper presents different algorithms and application possibilities under the paradigm of DNA bioinspired metaheuristics or algorithmics. The smart manufacturing frame is in my view an umbrella for the exposition, but the technologies of applicability can be considered separately. In my opinion, the paper can be better classified as a review or tutorial instead of an original research, since the exposition deals with existing algorithm applicability, with illustrative examples of the algorithm.

While originality in methodology cannot be claimed as a contribution, the tutorial introduction of those techniques is beyond ordinary review papers, in particular for techniques not widespread, so their panoramic overview is valuable.

Lines 92-97 include the scope of the presentation of the possibilities of DNA-RNA-Protein transfer of information in one direction and from simple path identification to complexity. Biomimetics is beyond the mere inspiration resemblance, so handling information from simple to aggregated combination transformed it into a natural algorithm itself.

This reviewer must declare himself incapable of fully following the full content of the examples but appreciates the step-by-step generative process the algorithms can offer, so giving potential value for ulterior consultation.

The examples are relevant in their technology field, so the different image processing purposes (similarity or segmentation), together with time series pattern recognition are important methodologies not only in manufacturing (smart) but in other fields.

The references are enough, but a more complete revision-like bibliography could improve the paper's value as a tutorial introduction to the potential of those techniques.

In my view, the conclusions putting DNA-based computing (DBC) in the core of the link between the records (past manufacturing) and present/future (ongoing manufacturing) is forced. While many metaheuristics can be applied to complex problems, the differential value of algorithms that generate solutions (emergency) from elementary combinations of information has a particular attractiveness for complex systems and with possibilities in cyber-physical systems linking both parts of the twin.     

Author Response

our answers are attached. thank you.
